# Navigating the Maze of Kinases: CaMK-like Family Protein Kinases and Their Role in Atherosclerosis

**DOI:** 10.3390/ijms25116213

**Published:** 2024-06-05

**Authors:** Jules T. J. Teuwen, Emiel P. C. van der Vorst, Sanne L. Maas

**Affiliations:** 1Institute for Molecular Cardiovascular Research (IMCAR), RWTH Aachen University, 52074 Aachen, Germany; jteuwen@ukaachen.de; 2Aachen-Maastricht Institute for CardioRenal Disease (AMICARE), RWTH Aachen University, 52074 Aachen, Germany; 3Interdisciplinary Center for Clinical Research (IZKF), RWTH Aachen University, 52074 Aachen, Germany; 4Institute for Cardiovascular Prevention (IPEK), Ludwig-Maximilians-Universität München, 80336 München, Germany

**Keywords:** kinases, CAMKL family, atherosclerosis, inflammation

## Abstract

Circulating low-density lipoprotein (LDL) levels are a major risk factor for cardiovascular diseases (CVD), and even though current treatment strategies focusing on lowering lipid levels are effective, CVD remains the primary cause of death worldwide. Atherosclerosis is the major cause of CVD and is a chronic inflammatory condition in which various cell types and protein kinases play a crucial role. However, the underlying mechanisms of atherosclerosis are not entirely understood yet. Notably, protein kinases are highly druggable targets and represent, therefore, a novel way to target atherosclerosis. In this review, the potential role of the calcium/calmodulin-dependent protein kinase-like (CaMKL) family and its role in atherosclerosis will be discussed. This family consists of 12 subfamilies, among which are the well-described and conserved liver kinase B1 (LKB1) and 5′ adenosine monophosphate-activated protein kinase (AMPK) subfamilies. Interestingly, LKB1 plays a key role and is considered a master kinase within the CaMKL family. It has been shown that LKB1 signaling leads to atheroprotective effects, while, for example, members of the microtubule affinity-regulating kinase (MARK) subfamily have been described to aggravate atherosclerosis development. These observations highlight the importance of studying kinases and their signaling pathways in atherosclerosis, bringing us a step closer to unraveling the underlying mechanisms of atherosclerosis.

## 1. Introduction

Cardiovascular diseases (CVD) are the leading cause of death globally and are responsible for one out of three deaths globally. Moreover, CVD is a major public health burden, and the number of people suffering from CVD globally has doubled since 1997 due to, amongst other factors, lifestyle changes and aging [1]. In addition to its impact on morbidity and mortality, the economic burden of CVD in countries of the European Union exceeds a total of EUR 210 billion and is expected to increase further [2]. Atherosclerosis is the main underlying cause of CVD, during which lipid-laden lesions (also known as plaques) develop in the arterial wall. Plaque formation occurs over a prolonged period, often years, without apparent symptoms [1,3,4]. Upon plaque progression, blood flow can be impeded or progressed atherosclerotic plaques may become unstable and rupture, followed by thrombus formation resulting in the occlusion of the vessel and thus, depending on the location, cause a stroke or myocardial infarction [1,4].

Mechanistically, blood flow is disturbed in atheroprone regions of arteries, activating the endothelial cells (ECs), which form the inner barrier of the blood vessel [5,6]. Activation of the endothelial layer results in an increase in its permeability [6,7]. This increased permeability promotes the relocation of low-density lipoprotein (LDL) particles and the accumulation of the particles in the intima, where they become oxidized [4,6,8,9]. Therefore, elevated LDL levels in the circulation are a major risk factor for atherosclerosis development [3,4]. Oxidized LDL (oxLDL) acts as a danger- or damage-associated molecular pattern and can interact with lectin-like oxLDL receptor 1 (LOX-1) on ECs, provoking an inflammatory response [3,10,11]. This response results in the secretion of cytokines and chemokines by the ECs. Additionally, ECs upregulate the expression of adhesion molecules on their cell surface, allowing immune cells, such as monocytes, to adhere to the endothelium and enter the vessel wall [4,12]. Within the vascular wall, these monocytes can differentiate into macrophages, which are capable of engulfing oxLDL particles [4,13]. This process leads to the formation of foam cells as macrophages are not able to metabolize oxLDL particles efficiently [14,15,16,17,18].

Moreover, vascular smooth muscle cells (VSMCs) migrate from the tunica media to the plaque surface, where they produce an extracellular matrix forming a fibrous cap, for example, containing collagen, that stabilizes the plaque [4,19]. Furthermore, VSMCs can also give rise to foam cells and can resemble a macrophage-like phenotype [4,20,21,22]. Notably, macrophages can produce matrix metalloproteinases (MMPs), a class of protease enzymes participating in the remodeling of the extracellular matrix. Degradation of the extracellular matrix by MMPs leads to a destabilized plaque, thus increasing the chance of rupture [4,23,24,25]. CD4^+^ T cells can influence this MMP release by secreting inflammatory cytokines, such as interferon-γ (IFN-γ) and interleukin-17 (IL-17) [4,26,27,28,29]. Eventually, the cells within the core of the plaque undergo apoptosis and necrosis, forming a so-called necrotic core, further promoting inflammation and plaque progression [4,24,30].

The underlying mechanisms of inflammation and thereby regulation of the vast number of cells that play a role in atherosclerosis are incredibly complex and involve many different molecular signaling cascades [23,28,31]. Adding to this complexity, first messengers (i.e., ligands) may induce different signaling cascades depending on the presence of their representative receptors on certain cell types [32]. Upon ligand binding, an intracellular signaling cascade involving protein kinases is activated. These protein kinases are a large group of enzymes, each playing a unique role in cell signaling with different specificities for target protein substrates [33]. By phosphorylating specific amino acid residues of their target proteins, these kinases can alter the stability, subcellular localization, and enzymatic activity of various proteins [32,33,34]. Interestingly, kinases are remarkably druggable targets, and many small molecules targeting kinases have been developed and are even already approved by regulatory agencies for clinical use, mostly, however, in the oncology field [35]. To date, no kinase inhibitors are available designated to target atherosclerosis. Therefore, in this review, we will provide further details on the kinome and particularly the calcium/calmodulin-dependent protein kinase (CaMK)-like (CaMKL) family and their role and target potential in atherosclerosis.

## 2. The Kinome

Protein kinases are classified into five categories. The categorization is based on the type of amino acid residue the kinases can phosphorylate. An Enzyme Commission (E.C.) number is assigned, resulting in distinct categories of protein kinases, namely protein-serine/threonine kinases (STK, E.C. 2.7.10), protein-tyrosine protein kinases (PTK, E.C. 2.7.11), dual-specificity kinases (i.e., acting on both serine/threonine and tyrosine residues, E.C. 2.7.12), protein-histidine kinases (E.C. 2.7.13), and protein-arginine kinases (E.C. 2.7.14) [36]. However, only the first three are relevant to humans, as the other two are more relevant to prokaryotic organisms [36,37,38,39]. Furthermore, only the kinases of the first two categories are studied more extensively [37]. Within these categories, several kinase groups, families, and subfamilies can be distinguished based on the sequence similarity between kinase domains (Figure 1) [40,41]. In this review, we will discuss the role of kinases and signaling pathways of the CaMKL family (Figure 1) in the context of atherosclerosis, and we will highlight which crucial information is still lacking in this field.

## 3. CaMK-like Family

In this review, we focus on the STK category, since the STK category is still rather unexplored and thereby a promising group to focus on to identify novel therapeutic targets to treat atherosclerosis. The PTK category has already been explored in more detail, and several drugs targeting kinases in this category have already been approved for clinical use [42]. The STK category consists of seven groups [40]. Some groups are well represented in the literature (for example, cyclin-dependent kinase (CDK), mitogen-activated protein kinase (MAPK), glycogen synthase kinase (GSK), CDK-like related (CMGC) group, and tyrosine kinase (TK) group) compared to, for example, the CaMK group [42,43]. The CaMK group is comprised of 18 families, and a notable family within the CaMK group is the CaMKL family, consisting of 12 subfamilies (Figure 1) [42,43]. A list of kinases that compromise this family and its subfamilies is provided in Table 1. The subfamilies and their potential role in atherosclerosis will be discussed in the chapters below.

### 3.1. The AMPK Subfamily in Atherosclerosis

The 5′-AMP-activated protein kinase (AMPK) subfamily consists of two members, which have an isoform-specific AMPK catalytic alpha subunit, namely α1 and α2, and are therefore known as AMPKα1 and AMPKα2. The alpha subunits form a heterotrimeric protein complex with a beta and a gamma subunit [44]. AMPK signaling promotes the generation of adenosine triphosphate (ATP) via catabolic pathways and enables metabolic reprogramming by, for example, modulating lipid metabolism and glycolysis [45,46,47]. Interestingly, immune cells depending on glycolysis are generally associated with pro-inflammatory pathways, while cells depending on oxidative phosphorylation are associated with anti-inflammatory pathways [46]. Therefore, AMPK signaling via liver kinase B1 (LKB1) activation has been suggested as a possible mechanism by which cells undergo metabolic reprogramming to acquire an anti-inflammatory phenotype [45,46]. For example, in vitro studies showed that AMPKα1 deficiency in bone marrow-derived macrophages (BMDM) and dendritic cells leads to an increase in pro-inflammatory cytokine release as well as increased expression of the costimulatory molecules CD80 and CD86 [48]. In addition, AMPK signaling has been reported to play an important role in the polarization of BMDMs to an M2 phenotype in vitro [49]. Moreover, activation of AMPK signaling in macrophages also reduced proliferation, inflammation, and lipid uptake in vitro [50,51,52,53]. In line with these observations, myeloid-specific deletion of *Ampkα1* or global knockout of *Ampkα2* promoted atherosclerosis development in LDL receptor^−/−^ (*Ldlr*^−/−^) mice [54,55]. Surprisingly, however, other investigators showed that global or myeloid-specific knockout of *Ampkα1* as well as myeloid-specific deletion of *Ampkα2* reduced plaque size by inhibiting monocyte-macrophage differentiation in apolipoprotein E^−/−^ (*Apoe*^−/−^) mice [56,57]. Even more striking is the observation that double knockout mice for both *Ampkα* subunits in myeloid cells showed no significant differences in myelopoiesis, plaque size, number of CD68^+^ lesional cells, or circulating lipids when atherosclerosis was induced by proprotein convertase subtilisin/kexin 9-AAV (PCSK9-AAV) injection [58]. In conclusion, while AMPK appears to act anti-inflammatory in in vitro experiments, there have been contradictory reports on the role of AMPK in atherosclerosis in vivo. A possible explanation for the contradictory in vivo data may be the use of different animal models by which atherosclerosis is induced (i.e., PCSK9-AAV injection, *Ldlr*^−/−^, or *Apoe*^−/−^ background), but a clear explanation remains to be investigated.

AMPK signaling has also been reported to promote differentiation of naïve T cells to T regulatory (T_reg_) cells [59,60]. This most likely occurs through the inhibition of mammalian target of rapamycin (mTOR) signaling and stimulation of fatty acid oxidation [61,62]. Since T_reg_ cells have been reported to play a crucial role in reducing inflammation and play a role in plaque regression, an increase in this T cell subset would be highly beneficial in the context of atherosclerosis [63].

In addition to these immune cells, AMPK signaling also plays a role in vascular cells, like ECs. In these cells, AMPK signaling reduces inflammatory signaling induced by tumor necrosis factor (TNF) and thereby decreases monocyte adhesion to the endothelium in vitro [64,65], suggesting an atheroprotective role for AMPK. A possible responsible mechanism could be the phosphorylation of p300 by AMPK, as this enzyme promotes nuclear factor kappa B (NF-κB) signaling [65]. Furthermore, AMPK signaling negatively regulates the production of reactive oxygen species (ROS) in mitochondria, which has been described as a potential mechanism for activation of the nucleotide-binding oligomerization domain (NOD)-, leucine-rich repeat (LRR)-, and pyrin domain-containing protein 3 (NLRP3) inflammasome activation in ECs [66,67,68,69,70,71]. The NLRP3 inflammasome is a supramolecular protein complex that can be formed in a variety of cell types and controls the bioactivity and release of IL-1β, promoting further inflammation [72,73]. Furthermore, the anti-inflammatory effect mediated by AMPK signaling in ECs has also been confirmed in the context of atherosclerosis in vivo, as the disturbed flow in atherosclerotic-prone regions has been reportedly associated with increased AMPK signaling, which led to enhanced glycolysis in ECs mediated by the transcription factor hypoxia-inducible factor 1α (HIF1α) [74].

AMPK signaling has also been described to play an important role in inhibiting lipid-induced secretion of inflammatory cytokines, lipid uptake, and CD68 expression in vitro in VSMCs [75,76]. CD68 expression is associated with a more macrophage-like phenotype, suggesting the occurrence of VSMC phenotype switching [77]. Furthermore, AMPK signaling destabilizes Runt-related transcription factor 2 (RUNX2) in VSMCs, which is characteristic of a more osteoblast-like phenotype, thereby protecting against atherosclerotic calcification [78]. Moreover, AMPK signaling reduces plaque instability by restraining the expression of Kruppel-like factor 4 (KLF4) [79,80]. This transcription factor plays a role in VSMC phenotype switching and has been reported to be upregulated in VSMCs found in unstable plaques [81]. The current data indicate that AMPK signaling in VSMCs inhibits phenotypic switching to a macrophage-like phenotype, reducing plaque instability and calcification, and therefore acting protectively in atherosclerosis.

Collectively, these results indicate an anti-inflammatory role for AMPK signaling across multiple cell types in vitro (Figure 2). However, in more complex in vivo models with myeloid or global knockout, conflicting results have been obtained so far, urging the need for further investigation.

### 3.2. Signaling by the BRSK, HUNK, and NIM1 Subfamilies in Atherosclerosis

To the best of our knowledge, there have been no studies focusing on the roles of brain-specific kinase 1 (BRSK1), BRSK2, hormonally up-regulated neu tumor-associated kinase (HUNK), or serine/threonine-protein kinase NIM1 (NIM1) in the context of inflammation or atherosclerosis so far.

### 3.3. CHK1 Subfamily and Its Role in Atherosclerosis

Checkpoint kinase 1 (CHK1, previously known as CHEK1) is the sole member of the CHK1 subfamily and is activated upon DNA damage, where it further has a role in the DNA damage response (DDR) pathway [82]. Mice lacking *Chk1* die early in embryogenesis because of a defective function of cell cycle checkpoints, and the cell nucleus displays morphologic abnormalities as early as the blastocyst stage [83]. Interestingly, COVID-19 has been shown to degrade CHK1, causing DNA damage due to a disturbed DDR, which is associated with the induction of pro-inflammatory pathways and cell senescence [84]. This is of interest since DNA damage, inflammation, and cell senescence are also associated with atherosclerosis [4,85,86,87].

Furthermore, stimulation of ECs with 7-ketocholesterol or lysophosphatidylcholine (components of oxLDL) has been associated with increased phosphorylation of CHK1, resulting in apoptosis and IL-8 production by ECs [88,89]. However, the exact role of CHK1 in these ECs has not been explored, and the potential role of CHK1 in atherosclerosis has yet to be discovered. Interestingly, other researchers reported that laminar flow over ECs induced phosphorylation of deSUMOylase sentrin-specific isopeptidase 2 (SENP2) by CHK1 [90]. The latter leads to decreased EC activation, reflected by a decreased expression of adhesion molecules, compared to cells experiencing a disturbed flow [90]. Strengthening this observation, SENP2 S344A knock-in mice, lacking the phosphorylation site of CHK1, had significantly larger lipid-laden lesions in the aorta when fed a high-fat diet, suggesting an atheroprotective role for CHK1 in ECs [90].

Contrary to the atheroprotective role of CHK1 in ECs, CHK1 might play an atherogenic role in VSMCs [91]. It has been observed that DNA damage induces the phosphorylation of CHK1, leading to a reduced expression of alpha smooth muscle actin (αSMA), indicating a loss of VSMC phenotype [91]. Notably, a CHK1 inhibitor (LY2603618, also known as Rabusertib) that had entered oncology clinical trials could rescue the expression of αSMA [91]. For this reason, future research should focus on studying the role of CHK1 signaling in more detail in atherosclerosis, and the focus could mainly be on ECs and the phenotypic switching of VSMCs.

### 3.4. The Role of the LKB1 Subfamily in Atherosclerosis

LKB1 is the sole member of the LKB1 subfamily and plays a role as a master kinase since it regulates the activities of many other protein kinases, among which the CaMKL subfamilies AMPK, microtubule affinity-regulating kinase (MARK), BRSK1, salt-inducible kinase (SIK), maternal embryonic leucine zipper kinase (MELK), novel (nua) kinase (NUAK), and sucrose non-fermenting-related kinase (SNRK) [92,93,94,95]. The essential role of LKB1 is also highlighted by the fact that a murine global knockout of *Lkb1* is embryonically lethal [96]. These mice developed multiple developmental defects, including neural tube defects, mesenchymal cell death, and vasculature abnormalities. Notably, these phenotypes were associated with tissue-specific deregulation of vascular endothelial growth factor (VEGF) expression [96]. Interestingly, overexpression of *Lkb1* delayed atherosclerosis development [97]. This observation might be explained by LKB1 regulating two processes reported to impede plaque development, namely suppression of mTOR signaling in an AMPK-dependent manner and Hippo kinase signaling by controlling the phosphorylation of MARKs [97,98,99,100,101,102,103]. Additionally, several studies have elucidated the role of LKB1 in different cells that play a key role in atherosclerotic plaques, like macrophages, T cells, and VSMCs. Therefore, these cell types and the role of LKB1 will be discussed below.

In lesional macrophages, the expression of LKB1 was downregulated when mice received a Western-type diet for 16 weeks, compared to mice fed a Western-type diet for 8 weeks. Moreover, in these lesional macrophages, scavenger receptor A (SRA) was reported to be degraded upon phosphorylation by LKB1. Consequently, deficiency of LKB1 in macrophages led to a high expression of SRA, resulting in an increased uptake of lipids, thereby promoting foam cell formation and atherosclerosis development in vivo [104]. Supporting this protective role of macrophage LKB1, others have reported that the regulation of autophagy by LKB1 in macrophages is another mechanism by which this kinase exerts atheroprotective effects. Autophagy in these cells leads to a reduced pro-inflammatory environment due to the reduction of inflammasome components. This effect is mediated by LKB1 signaling via AMPK, which causes the inhibition of mTOR [105,106]. Finally, LKB1 has been reported to inhibit NF-κB activation, a transcription factor regulating genes involved in the inflammatory response, possibly via a direct interaction with an inhibitor of κB (IκB) [107]. Overall, current research strongly points to the anti-inflammatory and atheroprotective role of macrophage LKB1.

In CD4^+^ T cells, LKB1 deficiency results in increased glycolysis, and this metabolic shift could be responsible for their preferential differentiation towards a T helper 1 (Th1) and Th17 phenotype [108,109]. Notably, these subsets of CD4^+^ T cells are associated with acute coronary syndrome [110]. In T cells, LKB1 has been reported to be essential for maintaining the expression of forkhead box protein P3 (FOXP3) [109]. FOXP3 is a master transcription factor in T_reg_ cells, a CD4^+^ T cell subset crucial to dampening inflammatory responses, and they have been reported to play a protective role in atherosclerosis [63,111,112]. Moreover, LKB1 signaling promotes the expression of a wide spectrum of immunosuppressive genes by a mechanism involving the augmentation of transforming growth factor-beta (TGF-β) signaling, a cytokine crucial to the differentiation of CD4^+^ T cells into T_reg_ cells [113,114]. Furthermore, the mevalonate pathway has been reported to be compromised in T_reg_ cells with LKB1 deficiency, and restoring this metabolic pathway promoted normal functioning of T_reg_ cells. Remarkably, the described LKB1-mediated regulation of T_reg_ cells was independent of AMPK [109]. However, another mechanism by which LKB1 is important to T_reg_ cells does depend on AMPK [115]. Collectively, these results indicate a crucial role for LKB1 signaling in reducing inflammation by undermining pro-inflammatory Th cell differentiation and maintaining normal T_reg_ function by controlling metabolic pathways.

LKB1 signaling also plays a role in lesional VSMCs, as LKB1 deficiency in these cells promoted atherosclerosis development in mice. Here, LKB1 was reported to bind to and phosphorylate sirtuin 6, thereby reducing LOX-1 expression and thus oxLDL uptake [116]. Interestingly, LKB1 has also been reported to inhibit the calcification of VSMC [117]. Taken together, LKB1 signaling appears to play a major anti-inflammatory role across a range of cell types and protects from atherosclerosis development (Figure 3).

### 3.5. MARK Subfamily and Its Relevance in Atherosclerosis

The MARK subfamily consists of four members: MARK1, MARK2, MARK3, and MARK4 [118]. MARK1, MARK3, and MARK4 have been reported to activate Hippo kinases, which may have implications for atherosclerosis development since activation of the Hippo kinase (mammalian STE20-like) MST1 has been reported to suppress EC activation when exposed to disturbed flow [98,102]. Although the literature on the role of MARKs in atherosclerosis is rather limited, there are some reports on MARK2 and MARK4. For example, a genome-wide association study (GWAS) revealed that rs10897458 in MARK2 is associated with circulating LDL levels [119]. However, further research is required to investigate the relevance and potential causal role of this single nucleotide polymorphism (SNP). Other researchers have reported that phosphorylation of MARK2 by LKB1 in a murine macrophage cell line (RAW264.7) plays a pro-inflammatory role characterized by the production of C-X-C motif ligand 15 (CXCL15), IL-1β, IL-6, and TNF [120].

Concerning MARK4, its expression has been reported to be increased in atherosclerotic lesions compared to adjacent areas [121]. Additionally, mice with a MARK4 hematopoietic deficiency had smaller atherosclerotic lesions [121]. Interestingly, in stainings of human atherosclerotic plaques, MARK4 colocalized with NLRP3 in macrophages [121]. Supporting this notion, in vitro studies with BMDMs that were deficient in MARK4 were protected against cholesterol crystal-induced NLRP3 inflammasome activation [121]. Cholesterol crystals and NLRP3 inflammasome activation are believed to promote inflammation and therefore play an important role in atherosclerosis [69,70,71]. However, the exact mechanisms by which MARK4 activates the NLRP3 inflammasome remain unclear. One possible mechanism could be that MARK4 activates c-Jun N-terminal kinase (JNK), a kinase that can phosphorylate NLRP3 at a site that is critical for deubiquitylation and inflammasome assembly [122,123]. Additionally, MARK4 has been reported to promote inflammation and mitochondrial oxidative stress via activating IκB kinase-α (IKKα), promoting NF-κB signaling [124,125]. Other proatherogenic effects exerted by MARK4 have been elaborately reviewed by Qin et al., and one effect they mentioned is the upregulation of the expression of the transcription factor sterol regulatory element-binding protein-1c (SREBP-1c), which promotes the expression of genes involved in lipogenesis in adipocytes [123,126,127]. Collectively, there is evidence pointing to a proatherogenic role for MARK4 (Figure 4), rendering this kinase a potential therapeutic target in atherosclerosis. Therefore, MARK4 inhibitors, such as MARK4 inhibitors 1–4 [128], may be worth exploring in an atherosclerosis model.

### 3.6. A Potential Role for the MELK Subfamily in Atherosclerosis

MELK is the sole member of the MELK subfamily and is involved in a variety of cellular processes, including carcinogenesis, apoptosis, and metabolism [129,130,131,132], but its exact function remains to be determined. Nevertheless, inhibition of MELK by OTSSP167 inhibited murine macrophage M1 polarization [133]. These findings suggest a pro-inflammatory role mediated by MELK signaling, which may have implications for atherosclerosis development, rendering MELK an interesting target for future studies.

### 3.7. Signaling by the NUAK Subfamily in Atherosclerosis

The NUAK family of SNF1-like kinase 1 (NUAK1, previously known as AMPK-related protein kinase 5 [ARK5]), together with NUAK2 (previously known as SNF1/AMP kinase-related kinase [SNARK]), form the NUAK subfamily. Most of the current knowledge regarding NUAK1 is derived from the field of oncology, where NUAK1 is known as a tumor survival factor that, amongst other functions, is involved in mitochondrial ATP production and mitochondrial dynamics (i.e., fission and fusion) [134,135]. Notably, in neurons, NUAK1 has very recently been reported to modulate both mitochondrial metabolism and trafficking [134]. Additionally, this kinase is involved in tumor invasion and metastasis of, for example, colorectal, pancreatic, and gastric cancers [135]. Interestingly, patients with acute coronary syndrome showed decreased expression of *NUAK1* in peripheral blood mononuclear cells (PBMCs) [136]. Unfortunately, the reason for and consequences of this decrease have not been investigated so far. Furthermore, in ovarian cancer cells, it has been reported that loss of signaling via LKB1-NUAK1 promotes NF-κB signaling [137]. Whether the latter observation holds true for cell types in the human plaque remains to be investigated. Considering the notion that restraining NF-κB signaling would dampen inflammatory signaling, this might indicate a possible protective role for NUAK1 in atherosclerosis. However, related to the scope of this review, to our knowledge, no information is available on the role of NUAK1 in atherosclerosis, and its precise role remains to be elucidated.

NUAK1’s sibling, NUAK2, plays a role in adipose tissue. An adipocyte-specific knockout of *Nuak2* caused inflammation of the white adipose tissue, ectopic lipid deposition in the liver and muscle, and impaired adaptive thermogenesis in the brown adipose tissue [138]. These data highlight an anti-inflammatory role for NUAK2 in the adipose tissue, which is of importance since adipocytes play an important role in atherosclerosis by releasing adipokines that regulate inflammation [138,139]. Furthermore, two SNPs in *NUAK2*, rs4682880 and rs4682676, were associated with body mass index, waist circumference, and an increased risk of obesity in women [138]. Obesity is an important risk factor for atherosclerosis and CVD [4,140], but the exact and potential causal role of these SNPs remains to be determined. Consistent with the anti-inflammatory role of NUAK2 in adipocytes, a similar role has been described in CD4^+^ T cells. More specifically, G protein-coupled receptor 65 (GPR65) expression has been linked to Th1/Th17 differentiation. Here, a negative correlation was found between *GPR65* and *NUAK2* expression, as *NUAK2* expression also increased upon a CD4^+^ T cell-specific knockout of *GPR65*. Interestingly, silencing of *NUAK2* in these GPR65-deficient CD4^+^ T cells restored Th1/Th17 differentiation, indicating an anti-inflammatory role for NUAK2 [141]. Together, this indicates an anti-inflammatory role for NUAK2 signaling in adipocytes and CD4^+^ T cells, but a study in the context of atherosclerosis is lacking so far.

### 3.8. Modulating Lipid Metabolism by the PASK Subfamily in Atherosclerosis

Per-arnt-sim (PAS) domain-containing serine/threonine-protein kinase (PASK) is the only member of the PASK subfamily and has been shown to control the release of the pancreatic islet hormone and insulin sensitivity [142,143]. Furthermore, PASK can activate SREBP-1, resulting in an increased expression of genes involved in cholesterol metabolism [127,144]. Interestingly, knockdown of *PASK* in hepatocellular carcinoma cells (HepG2) inhibited lipid accumulation, oxidative stress, and the release of inflammatory cytokines [145]. These findings indicate a role for PASK in modulating lipid metabolism, which may, in turn, impact atherogenesis.

### 3.9. The SIK Subfamily in Atherosclerosis

The SIK subfamily consists of SIK1, SIK2 (previously known as QIK), and SIK3 (previously known as QSK). SIK1 expression has been reported to be induced by TGFβ signaling and, in turn, negatively regulates the activity of this pathway, thereby decreasing inflammation [114,146]. Moreover, an important role of both SIK1 and SIK2 is the suppression of the function of the transcription factor cAMP response element-binding protein (CREB) by phosphorylating and inactivating transducer of regulated CREB-binding proteins (TORC) [147,148]. Interestingly, CREB has been reported to enhance the production of IL-17A by peritoneal macrophages and promote inflammation in an atherosclerosis mouse model [149]. Therefore, SIK-mediated inhibition of CREB may reduce inflammation in atherosclerosis. SIK1 was reported to be a key factor in regulating the stability of intercellular junctions in epithelial cells by controlling the expression of E-cadherin via CREB inhibition [150]. Furthermore, SIK1 deficiency leads to decreased expression of E-cadherin as well as reduced transepithelial electrical resistance, indicating the epithelial junctions are less stable [150]. Notably, E-cadherin is also known to be expressed by ECs [7], but a similar role for SIK1 in this cell type remains to be validated. If SIK1 functions in a similar way in ECs, SIK1 signaling could be atheroprotective by improving the barrier function of the endothelium; however, this remains to be explored.

Furthermore, SIK2 as well as SIK3 play a role in macrophage polarization, as their inhibition promoted the transition to an anti-inflammatory M2-like macrophage phenotype with enhanced production of IL-10 and reduced secretion of inflammatory cytokines [151,152]. These macrophages are generally considered to play a protective role in the context of atherosclerosis [153]. Moreover, the inhibition of SIKs in dendritic cells also promotes their immunoregulatory function by increasing IL-10 secretion as well as reducing inflammatory cytokine release [154]. Additionally, SIK3 has also been reported to positively regulate mTOR signaling [103,155], which may also be detrimental in the context of atherosclerosis.

In summary, current evidence points to a pro-inflammatory role for the kinases of the SIK subfamily in macrophages and dendritic cells since their inhibition promotes M2 polarization of macrophages and reduces pro-inflammatory cytokine release by macrophages and dendritic cells, which makes them promising targets to further explore in the context of atherosclerosis.

### 3.10. The SNRK Subfamily in Atherosclerosis

Within the SNF-related serine/threonine-protein kinase (SNRK) subfamily, SNRK is the only member, and it plays a role in insulin resistance [156]. The expression of SNRK has been reported to be decreased in the adipocytes of obese mice compared to lean mice [157]. Notably, inflammatory signals, more specifically treatment with TNF or overexpression of constitutively active IKKβ, also decreased SNRK expression in vitro in adipocytes [157]. Knockdown of *SNRK* in adipocytes enhanced phosphorylation of JNK and IKKβ, promoting lipolysis and expression of chemokines [157], which suggests an anti-inflammatory function of SNRK in adipocytes. A similar anti-inflammatory role has been described in cardiomyocytes since the cardiomyocyte-specific knockout of *Snrk* resulted in increased NF-κB signaling, increased macrophage infiltration into the heart, and the dysfunction of the left ventricle following angiotensin II infusion [158]. A possible underlying mechanism that could explain the previous anti-inflammatory observation is that SNRK can directly interact with NF-κB/p65 and mediate its anti-inflammatory effects [159]. One notable observation is that this interaction occurs independently of SNRK kinase activity, as a kinase-dead mutant form of SNRK was still able to interact with p65 [159]. Collectively, these reports indicate an anti-inflammatory role for SNRK in a variety of cell types, with a prominent role in dampening NF-κB signaling, which may have repercussions for the development of atherosclerosis.

## 4. From Bench to Bedside: Modulating CaMKL Family Kinases in the Clinic

Some compounds have been described as indirectly modulating the kinase activity of CaMKL family members. For instance, bempedoic acid (also known as ETC-1002) is an approved drug for treating hypercholesterolemia and atherosclerosis [160]. Interestingly, this drug has been reported to regulate leukocyte homing, adipose tissue inflammation in mouse models, and pro-inflammatory cytokine and chemokine release in human macrophages via the LKB1-AMPK axis [161]. Another drug, metformin, has a long history (>60 years in the clinic) as a first-line medication commonly used for the treatment of type 2 diabetes, and repurposing Metformin for additional applications has received increased attention recently, as reviewed by Foretz et al. [162]. Notably, metformin has also been demonstrated to promote AMPK signaling in macrophages and VSMCs and exert beneficial effects in the context of atherosclerosis [51,76,163]. As reviewed above, the LKB1-AMPK axis has been described as an atheroprotective signaling pathway across multiple cell types. Thus, these drugs might benefit patients suffering from atherosclerosis. Although evidence exists of beneficial effects on cardiovascular outcomes [164], robust evidence of metformin’s efficacy for cardiovascular outcomes is lacking. This is because the pre- and post-marketing cardiovascular outcome trials (that have been mandated since 2008 by the FDA for all new antidiabetic drugs), specifically exploring CVD endpoints independent of glucose-lowering effects, have not yet been performed for this drug.

The previously mentioned compounds promote signaling via kinases; however, some inhibitors of CaMKL family members are already being tested in clinical trials. Notably, several CHK1 inhibitors (Prexasertib (LY2606368), SCH 900776, BBI-355 (trial identifier: NCT05827614), and PEP07 (trial identifier: NCT05659732 and NCT05983523)) are in phase 1/2 of a clinical trial studying oncotherapy [165,166,167]. Moreover, the SIK2/3 inhibitor GRN-300 is in phase 1 studies and focuses on gynecological cancers [168]. Finally, a phase 1 breast cancer study involving the MELK inhibitor OTS167PO is currently underway (trial identifier: NCT02926690). However, rigorous evidence of whether these kinase inhibitors also modulate cardiovascular outcomes remains to be determined.

## 5. Conclusions and Future Perspectives

Most of the current treatment strategies for atherosclerosis focus on lowering circulating LDL levels. However, while these treatments are effective, CVD remains the leading cause of death worldwide, highlighting the importance of developing new treatment strategies. A step in this direction would be to obtain a better understanding of the signaling pathways involved in atherosclerosis development. Providing novel targets involved in these signaling pathways might provide the opportunity to address this global health burden by obviating the progression of atherosclerotic plaque. In addition, by targeting protein kinases instead of receptors or ligands, one can modulate the activity of multiple pathways that are underlying the disease. Targeting protein kinases may, therefore, be an advantage to consider since often more than one pathway participates in the disease. As described in detail above, LKB1 signaling across a multitude of cell types generally seems to act in an atheroprotective way and dampens inflammatory signaling pathways. For this reason, testing LKB1 agonists may be promising to target atherosclerosis. The same applies to the AMPK subfamily, although more research may be necessary to understand their role in vivo.

Another challenge will be to investigate in which signaling pathways the CaMKL family is involved. This is of importance since CaMKL family protein kinases play a fundamental role in mediating inflammation and atherogenic processes (Table 2). However, studies with a particular focus on atherosclerosis are lacking and are desired to gain a better fundamental understanding of the role of these kinases in the disease. A particular subfamily of interest targeting atherosclerosis is the SIK subfamily. There is growing evidence that their inhibition in immune cells renders an anti-inflammatory immunoregulatory phenotype, which makes them a promising therapeutic target. The challenge, however, would be to avoid off-target effects and find a way to specifically target the protein kinases of interest within the immune cell. For this reason, novel drug delivery methods should be developed to specifically target protein kinases in cell subsets. Finally, given the evidence that PASK influences lipid metabolism in liver cells, it may be worth exploring further whether this also holds true for other cell types to uncover a more specific role in the context of atherosclerosis. Altogether, a better understanding of CaMKL family protein kinases and the way they modulate disease pathways will pave the way to discovering novel strategies to combat atherosclerosis.

## Figures and Tables

**Figure 1 ijms-25-06213-f001:**
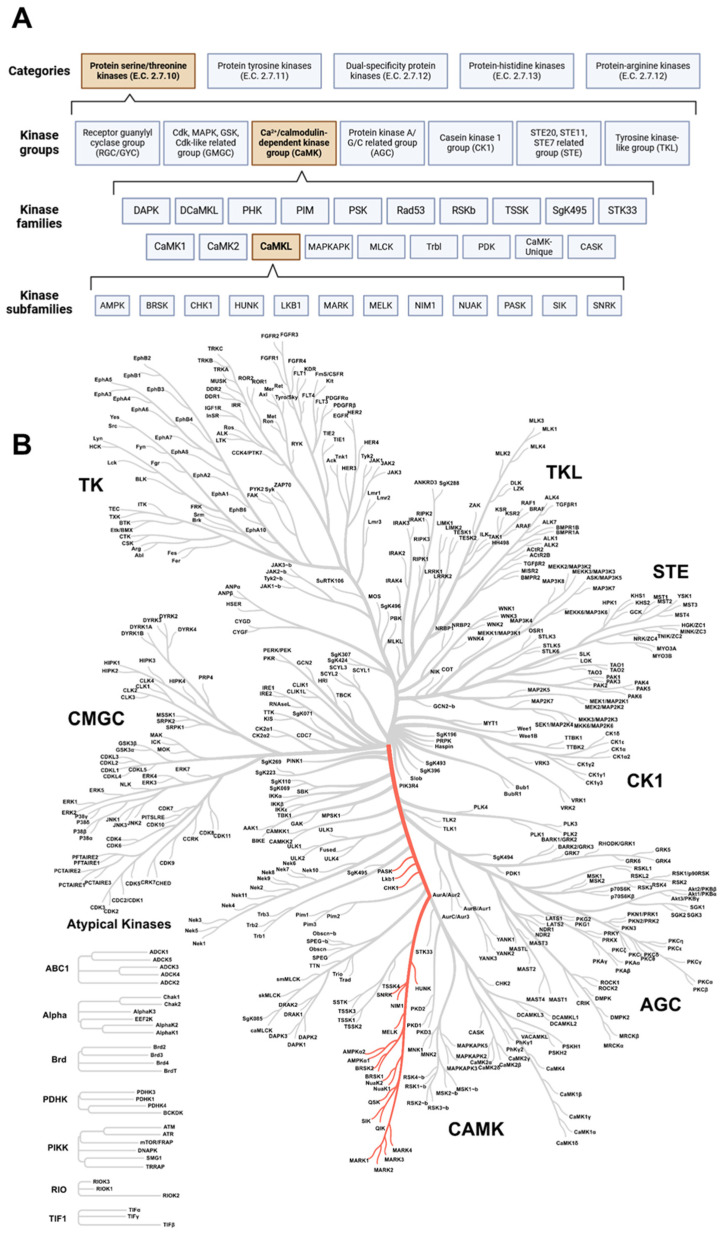
**The maze of protein kinases**. (**A**) Classification of protein kinases according to their amino acid residue phosphorylate capacity. Within the categories, several kinase groups, families, and subfamilies can be distinguished; subdivision is based on sequence similarity between protein kinase domains. The CaMKL family (part of the CaMK group) is the focus of this review and is highlighted in orange. Created with BioRender.com (accessed on 3 June 2024). (**B**) Coral tree showing the degree of sequence similarity between protein kinases. The location of the CaMK-like family is highlighted in red. CaMK: calcium/calmodulin-dependent protein kinase group; CaMKL: calcium/calmodulin-dependent protein kinase-like family. Created with phanstiel-lab.med.unc.edu/CORAL/ (accessed on 3 June 2024).

**Figure 2 ijms-25-06213-f002:**
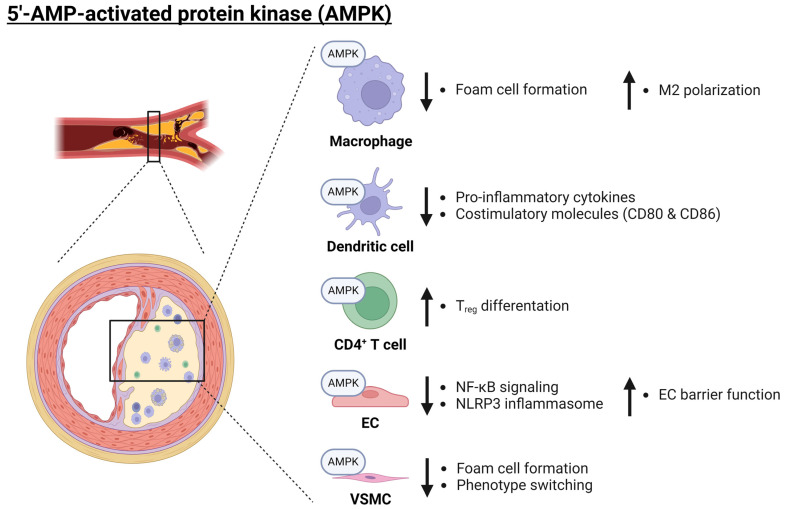
**Role of AMPK in atherosclerosis**. AMPK plays an anti-inflammatory role across a variety of cell types (macrophages, dendritic cells, CD4^+^ T cells, ECs, and VSCMCs) and thereby protects against the development of atherosclerosis. AMPK inhibits foam cell formation and promotes M2 polarization in macrophages. Additionally, AMPK reduces pro-inflammatory cytokine release and decreases the expression of costimulatory molecules in dendritic cells. Furthermore, AMPK has an anti-inflammatory role within ECs, where it inhibits NF-κB signaling as well as NLRP3 inflammasome activation whilst enhancing the barrier function of the endothelium. Finally, AMPK may promote T_reg_ differentiation as well as inhibit VSMC phenotype switching. Arrows pointing up and down indicate an increase or decrease, respectively. AMPK: 5′-AMP-activated protein kinase catalytic; EC: endothelial cell; M2: alternatively activated macrophage; NF-κB: nuclear factor kappa B; NLRP3: nucleotide-binding oligomerization domain (NOD)-, leucine-rich repeat (LRR)-, and pyrin domain-containing protein 3; T_reg_: regulatory T cell; VSMC: vascular smooth muscle cell. Created with BioRender.com (accessed on 3 June 2024).

**Figure 3 ijms-25-06213-f003:**
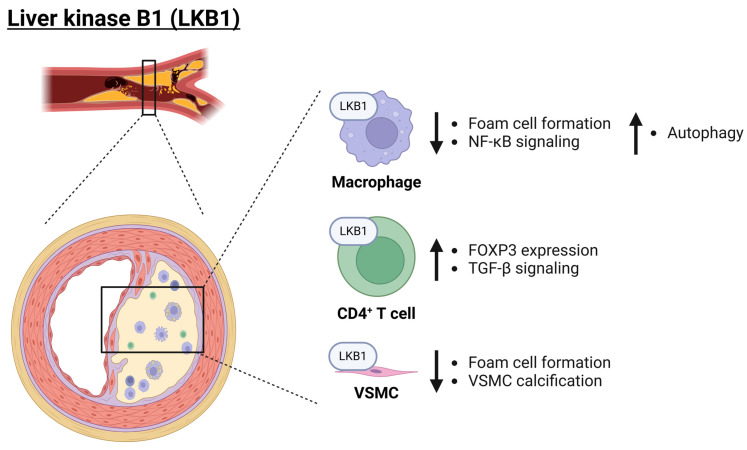
**Role of LKB1 in atherosclerosis**. LKB1 plays a major anti-inflammatory role across a variety of cell types (macrophages, CD4^+^ T cells, and VSMCs) and thereby protects from atherosclerosis development. Importantly, LKB1 inhibits foam cell formation but also inhibits NF-κB signaling as well as promotes autophagy of inflammasome components. Moreover, in CD4^+^ T cells, it promotes *FOXP3* expression and TGF-β signaling. Finally, in VSMCs, LKB1 reduces vascular calcification. Arrows pointing up and down indicate an increase or decrease, respectively. EC: endothelial cell; FOXP3: forkhead box protein P3; LKB1: liver kinase B1; NF-κB: nuclear factor kappa B; TGF-β: transforming growth factor-beta; VSMC: vascular smooth muscle cell. Created with BioRender.com (accessed on 3 June 2024).

**Figure 4 ijms-25-06213-f004:**
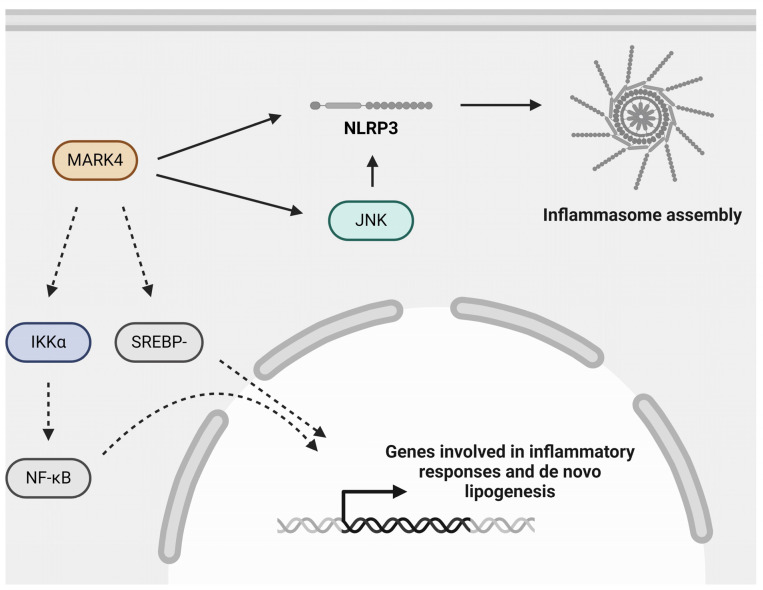
**Role of MARK4 in atherogenic processes**. MARK4 may aggravate atherosclerosis by regulating inflammasome activation, possibly via activating JNK, NF-κB signaling via its upstream kinase IKKα, and SREBP-1c signaling. Inflammasome activation and NF-κB signaling are important mediators of inflammation and are detrimental in the context of atherosclerosis. SREBP-1c signaling is involved in de novo lipogenesis. IKKα: inhibitor of κB (IκB) kinase-α; JNK: c-Jun N-terminal kinase; MARK4: microtubule affinity-regulating kinase 4; NF-κB: nuclear factor kappa B; NLRP3: nucleotide-binding oligomerization domain (NOD)-, leucine-rich repeat (LRR)- and pyrin domain-containing protein 3; SREBP-1c: sterol regulatory element-binding protein-1c. Created with BioRender.com (accessed on 3 June 2024).

**Table 1 ijms-25-06213-t001:** Overview of the CaMKL family of protein kinases.

Subfamily	Name	Kinase Name	UniprotID
AMPK	AMPKa1	5′-AMP-activated protein kinase catalytic subunit alpha-1	Q13131
	AMPKa2	5′-AMP-activated protein kinase catalytic subunit alpha-2	P54646
BRSK	BRSK1	Brain-specific kinase 1	Q8TDC3
	BRSK2	Brain-specific kinase 2	Q8IWQ3
CHK1	CHK1	Checkpoint kinase 1	O14757
HUNK	HUNK	Hormonally up-regulated neu tumor-associated kinase	P57058
LKB	LKB1	Liver kinase B1	Q15831
MARK	MARK1	Microtubule affinity-regulating kinase 1	Q9P0L2
	MARK2	Microtubule affinity-regulating kinase 2	Q7KZI7
	MARK3	Microtubule affinity-regulating kinase 3	P27448
	MARK4	Microtubule affinity-regulating kinase 4	Q96L34
MELK	MELK	Maternal embryonic leucine zipper kinase	Q14680
NIM1	NIM1	Serine/threonine-protein kinase NIM1	Q8IY84
NUAK	NUAK1	NUAK family of SNF1-like kinase 1	O60285
	NUAK2	NUAK family of SNF1-like kinase 2	Q9H093
PASK	PASK	PAS domain-containing serine/threonine-protein kinase	Q96RG2
SIK	SIK1	Salt-inducible kinase 1	P57059
	SIK2	Salt-inducible kinase 2	Q9H0K1
	SIK3	Salt-inducible kinase 3	Q9Y2K2
SNRK	SNRK	Sucrose non-fermenting-related kinase	Q9NRH2

**Table 2 ijms-25-06213-t002:** Overview of the functional roles of CaMKL family protein kinases.

Subfamily	Kinase	Cell Type	Functional Role	References
AMPK	AMPKα1	Macrophage and dendritic cell	Regulation of inflammatory responses	[48]
		Macrophage	Increased M2 macrophage polarization	[49]
			Inhibition of foam cell formation	[50,51,52,53]
		CD4^+^ T cell	Promotion of T_reg_ differentiation	[59,60,61]
		EC	Inhibition of NF-κB signaling	[64,65]
			Inhibition of NLRP3 inflammasome activation	[66]
			Enhancement of EC barrier function	[74]
		VSMC	Inhibition of foam cell formation	[75]
			Inhibition of phenotype switching	[78,80]
	AMPKα2	Macrophage	Increased M2 macrophage polarization	[49]
			Inhibition of foam cell formation	[51,52,53]
		CD4^+^ T cell	Promotion of T_reg_ differentiation	[59,60,61]
		VSMC	Inhibition of phenotype switching	[53,80]
CHK1	CHK1	EC	Reduced EC activation	[90]
		VSMC	Regulation of αSMA expression	[91]
LKB1	LKB1	Macrophage	Inhibition of foam cell formation	[104]
			Increased autophagy	[105,106]
			Inhibition of NF-κB signaling	[107]
		CD4^+^ T cell	Maintenance of FOXP3 expression	[108,109,115]
			Augmentation of TGF-β signaling	[113]
		VSMC	Inhibition of foam cell formation	[116]
			Inhibition of VSMC calcification	[117]
MARK	MARK1	HEK293T, LS174T, DLD1, MCF7, HaCaT	Activation of Hippo kinases	[98]
	MARK2	RAW264.7	Increased inflammation	[120]
	MARK3	HEK293T, LS174T, DLD1, MCF7, HaCaT	Activation of Hippo kinases	[98]
	MARK4	HEK293T, LS174T, DLD1, MCF7, HaCaT	Activation of Hippo kinases	[98]
		Macrophage	Regulation of the NLRP3 inflammasome	[121,122,123]
		Adipocyte	Promotion of NF-κB signaling	[124,125]
		3T3-L1	Regulation of SREBP-1c	[123]
MELK	MELK	Macrophage	Increased M2 macrophage polarization	[133]
NUAK	NUAK1	Primary neuronal cells, KMS-11, Sachi, HeLa, HCT116 p53-null, MCF-7	Regulation of mitochondrial dynamics and metabolism	[134,169,170]
		OVCAR8, FT190	Regulation of NF-κB signaling	[137]
	NUAK2	CD4^+^ T cell	Restraining Th1/Th17 differentiation	[141]
PASK	PASK	HepG2	Regulation of lipid metabolism	[145]
SIK	SIK1	HaCaT	Regulation of TGF-β signaling	[146]
		Y1	Inactivation of TORC	[147]
		HK2	Regulation of intracellular junctions	[150]
		Dendritic cell	Promotion of immunoregulatory function	[154]
	SIK2	HEK293T, MIN6	Inactivation of TORC	[148]
		Dendritic cell	Promotion of immunoregulatory function	[154]
		Macrophage	Increased M1 macrophage polarization	[151,152]
	SIK3	Dermal fibroblast	Stimulate mTOR signaling	[155]
		Dendritic cell	Promotion of immunoregulatory function	[154]
		Macrophage	Increased M1 macrophage polarization	[151,152]
SNRK	SNRK	Glomerular EC, HL-1, 3T3-L1	Inhibition of NF-κB signaling	[157,158,159]

## Data Availability

Not applicable.

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
