# Peer review of "Navigating the Maze of Kinases: CaMK-like Family Protein Kinases and Their Role in Atherosclerosis"

_ijms, 2024, doi:10.3390/ijms25116213_

Round 1

Reviewer 1 Report

Comments and Suggestions for Authors

The review is well-written and discussed a current hot, yet newly emerging topic in the atherosclerosis field.  Reviewer comments, mostly minor, are listed below:

*Fig 1B is too small to be comprehensible.  If font size is an issue within the main text, then this should be shuffled into a supplementary materials section instead.

*A stronger rationale to focus on the STK category should be explicitly stated.

*What do the authors mean by "full bock" knockout?  Does this indicate a "global" KO?  This term is not utilized within scientific literture, so please clarify.

*Please provide more text in the figure legends for Figures 2-4 to guide the reader through the illustrations.

*If possible, please use different types of images for Figures 2 and 3, to better distinguish different topics are discussed between these two figures.

*There are abbreviations used throughout which some audiences may not be familar with, so please spell-out accordingly when introducing more unfamilar abbreviations.

*My main conflict with the review is the function of SREBP-1c being shown in Figure 4.  SREBP-2 is the transcription factor involved in cholesterol biosynthesis, while SREBP-1c is involved in de novo lipogenesis.  If the authors have a strong enough rationale to want to retain the "1c" isoform, then at the very least the figure should only include "SREBP" so that all 3 isoforms (1a, 1c, 2) can be clumped together.

Comments on the Quality of English Language

English is fine.

Reviewer 2 Report

Comments and Suggestions for Authors

<Comments>

Line 30: "Cardiovascular disease (CVD) is the leading cause of death worldwide and represents a major public health burden [1]" was described. However, please be more specific. For example, how many people in the world have CVD today? And while CVD is the leading cause of death worldwide, what percentage of deaths are caused by CVD? Since this manuscript is a Review, the social context of CVD should also be described in more detail.

Reviewer 3 Report

Comments and Suggestions for Authors

Teuwen and colleagues wrote a beautiful review on the roles of the CaMK-like kinase family in atherosclerosis. The review is comprehensive and well-organized. I particularly like the schematics and summary tables, which make it much easier for the readers to identify useful information. I don’t have any more specific suggestions for the publication.

Round 2

Reviewer 2 Report

Comments and Suggestions for Authors

I am satisfied with the authors' response.